# Persistence of the ground beetle (Coleoptera: Carabidae) microbiome to diet manipulation

Anita Silver[1], Sean Perez[1], Melanie Gee[1], Bethany Xu[1], Shreeya Garg[1], Kipling Will[1,2]*, Aman Gill[1]

**1** Department of Environmental Science, Policy and Management, University of California Berkeley, Berkeley, California, United States of America, **2** Essig Museum of Entomology, University of California Berkeley, Berkeley, California, United States of America

* kipwill@berkeley.edu

**Data Availability Statement:** All data files are available via the NCBI Sequence Read Archive (BioProject PRJNA703093).

**Funding:** KW. NSF DEB #1556957. National Science Foundation Division of Environmental

## Abstract

Host-associated microbiomes can play important roles in the ecology and evolution of their insect hosts, but bacterial diversity in many insect groups remains poorly understood. Here we examine the relationship between host environment, host traits, and microbial diversity in three species in the ground beetle family (Coleoptera: Carabidae), a group of roughly 40,000 species that synthesize a wide diversity of defensive compounds. This study used 16S amplicon sequencing to profile three species that are phylogenetically distantly related, trophically distinct, and whose defensive chemical secretions differ: *Anisodactylus similis* LeConte, 1851, *Pterostichus serripes* (LeConte, 1875), and *Brachinus elongatulus* Chaudoir, 1876. Wild-caught beetles were compared to individuals maintained in the lab for two weeks on carnivorous, herbivorous, or starvation diets (n = 3 beetles for each species-diet combination). Metagenomic samples from two highly active tissue types—guts, and pygidial gland secretory cells (which produce defensive compounds)—were processed and sequenced separately from those of the remaining body. Bacterial composition and diversity of these ground beetles were largely resilient to controlled changes to host diet. Different tissues within the same beetle harbor unique microbial communities, and secretory cells in particular were remarkably similar across species. We also found that these three carabid species have patterns of microbial diversity similar to those previously found in carabid beetles. These results provide a baseline for future studies of the role of microbes in the diversification of carabids.

## Introduction

Insects are by far the most diverse group of animals [1, 2], and it is becoming clear that the success of several major insect groups is due in part to their resident microbiomes [2, 3]. However, microbiomes remain understudied in many major groups of insects, including Carabidae ground beetles. Carabidae consists of around 40,000 described species, making it one of the most species-rich animal families on earth [4]. Moreover, the variety of defensive chemicals produced in the carabid pygidial gland system is an impressive example of evolutionary

Biology. https://www.nsf.gov/bio/deb/about.jsp The funders had no role in study design, data collection and analysis, decision to publish, or preparation of the manuscript.

**Competing interests:** The authors have declared that no competing interests exist.

diversification [5]. Secretory cells of the pygidial gland system produce such diverse classes of molecules as carboxylic acids, formic acid, quinones, hydrocarbons, and aromatics; chemical diversity exists even within some genera [5]. Whether microbes play a functional role in carabid chemical diversity has not yet been studied.

Interactions between insects and their associated microbiomes can contribute to insect diversification [3]. Microbiomes can benefit host insects in many ways, such as producing vitamin B [6], regulating host metabolism in response to stress [7], and contributing to host development [8]. Notable examples of microbial symbionts supporting nutrient acquisition in insects include *Buchnera* bacteria producing essential amino acids allowing aphids to live on a nutrient-poor diet [9] and highly diverse termite gut microbes digesting cellulose for their wood-feeding hosts [10, 11]. Unlike aphids and termites, carabids tend to be dietary generalists, but microbial species are also known to contribute to other host phenotypes, including nutrient acquisition and detoxification. In ants [12], *Harpalus pensylvanicus* (Degeer, 1774) (Carabidae) [13] and *Cephaloleia* (Coleoptera: Chrysomelidae) [14], microbial symbionts assist their hosts in metabolizing different food sources. It is known that bacterial symbionts enable several beetle species to thrive in chemically hostile environments. For example, the mountain pine beetle *Dendroctonus ponderosae* (Hopkins, 1902) (Coleoptera: Curculionidae) can inhabit pine trees because its microbes break down defensive terpenes produced by the trees [15]. The microbiomes of *Nicrophorus vespilloides* Herbst, 1783 (Coleoptera: Silphidae) and other carrion beetles protect their hosts from toxins and speed up host digestion, making it easier for these beetles to feed on decaying carcasses [16, 17]. Insects are well known to benefit from defensive and protective symbioses. *Lagria villosa* (F.) (Coleoptera: Tenebrionidae) beetles live in symbiosis with *Burkholderia gladioli* that protect their host's eggs from pathogens by producing the antifungal compound lagriamide [18]. *Paederus* (Coleoptera: Staphylinidae) beetles are well known for producing toxic hemolymph that causes severe dermatitis; the toxin, pederin, is produced by a *Pseudomonas*-like symbiont [19]. The Asian citrus psyllid *Diaphorina citri* (Kuwafyama, 1908) (Hemiptera: Liviidae), an invasive pest in the U.S. that causes citrus disease, harbors endosymbiotic *Candidatus* Profftella armatura (Betaproteobacteria) that produce diaphorin, a toxin similar to pederin [20].

In carabid beetles, pygidial gland secretory cells perform an important metabolic function by synthesizing defensive chemicals for secretion. To our knowledge, no previous studies have compared the microbial communities of secretory cells from different carabid species. Given the multitude of established insect-bacterial associations as well as the diversity of ground beetle defensive chemistry, it is worth investigating the microbial diversity of secretory cell tissues as a first step in exploring potential functional links between secretory cell microbes and carabid defensive chemical biosynthesis.

If bacteria have contributed to the diversification of carabid beetle phenotypes, this connection may be reflected in patterns of carabid microbiome composition and diversity. Insect microbiome composition can be explained by several factors, such as host phylogeny [2, 21–23], dietary guild [23] or sampling locality [24]. Although many insects have persistent host-associated communities, some do not, highlighting the potential for fluctuations in microbial diversity; for example, some lepidoptera caterpillar microbiomes consist entirely of microbes ingested with leaves, with constant turnover based on short-term diet [25]. If the carabid microbiome is characterized by rapid, near-complete compositional turnover similar to that of the caterpillar microbiome, then it would be sensitive to dietary shifts [12, 26] or other changes to the local environment [11], and not obviously correlate with factors such as host phylogeny, chemistry, or tissue type. The composition of persistent host-associated microbiomes can also be influenced by changes to host diet, as has previously been found in some Coleoptera and

Lepidoptera species [14, 27], but would not exhibit near-complete compositional turnover as a result of these short-term perturbations.

As insects have an open circulatory system that allows hemolymph to flow throughout the body, microbial communities are found in many insect tissues [28]; but as in other animals, insects often have distinct microbial communities in different tissues [10, 12, 28]. Some of this diversity may relate to the variety of conditions found within insect anatomy, including aerobic and anaerobic regions and extreme pH gradients [11, 28]. Tissue-specific diversity could also be explained by a co-evolutionary relationship between hosts and symbiotic microbiota, in that hosts can harbor functionally useful bacteria in specialized tissues. For example, termites regulate unique microbiomes in each of several gut pouches [11], and many insect species maintain useful symbionts in specialized cells called bacteriocytes [28]. The present study focuses on the pygidial gland secretory cells (hereafter simply "secretory cells") and the gut. We focus on the microbial communities of these tissues because they are responsible for defensive chemical synthesis and digestion of food respectively—metabolic processes known to involve bacterial symbionts in other insect taxa.

In this study, we used 16S metagenomic amplicon sequencing to quantify the bacterial diversity hosted by three carabid species under several dietary treatments. Each host species produces distinct primary defensive compounds: *Anisodactylus similis* produces formic acid, *Pterostichus serripes* carboxylic acids, and *Brachinus elongatulus* quinones [29] [Will & Attygalle, unpublished data]. *Anisodactylus similus* has a distinct natural feeding preference from the other two species, so together these three species represent two different trophic types. *Brachinus elongatulus* and *P. serripes* are naturally generalist predator-scavengers, preferring animal matter but observed in nature and in the lab to eat a wide variety of sugar and protein rich plant and animal material; in contrast, *A. similis* is typically observed feeding on fallen fruits, seeds, and pollen [30] [Will unpbl.]. In addition to sequencing wild-caught beetles preserved at the time of collection, we also subjected live beetles of each species to three controlled dietary treatments.

This preliminary study was intended to reveal how short-term dietary shifts in beetle hosts affect bacterial diversity and composition, providing an initial step in broader efforts to characterize carabid microbial diversity in relation to host diversification. If carabids lack established microbiomes altogether, as seen in caterpillars [25], then diet treatment alone might explain a significant amount of the variation across microbial communities. In that case, relative to conditions observed in the field, same-host communities under different diet treatments might diverge, and different-host communities under the same diet treatment might converge. On the other hand, if carabids have resilient microbial communities—structured by environmental exposure at earlier life stages, vertical transmission, and/or symbiosis—we would expect those communities to remain largely intact during controlled changes to host diet. Further, in that case we expect that community diversity and composition would correlate with host characteristics such as species, tissue type, or defensive chemistry. To better distinguish these possible associations, our study quantifies the effect of dietary shifts on several host species and tissues separately. Notably, to our knowledge this study is the first to compare microbial communities of carabid pygidial gland secretory cells from multiple species. This work provides a baseline for future studies to investigate the connection between host-associated microbes and carabid beetle phylogenetic and chemical diversity.

## Methods

### Beetle husbandry and dissection

Twelve individuals each of *Anisodactylus similis* LeConte, 1851, *Pterostichus serripes* (LeConte, 1875), and *Brachinus elongatulus* Chaudoir, 1876 were collected (total 36 specimens).

*Pterostichus serripes* and *A. similus* were collected from U.C. Berkeley's Whitaker's Forest, Tulare County, CA (36.7022˚, -118.933˚). *Brachinus elongatulus* were collected from national forest land in Madera Canyon, Santa Cruz County, AZ (31.72˚, -110.88˚). The number of replicate beetles used for this study was constrained by the practical difficulties of finding and catching sufficient numbers of live specimens from multiple species. For each species, three wild-caught specimens were preserved in 95% ethanol immediately upon collection, and the remaining beetles were transported live to laboratory facilities on the U.C. Berkeley campus. For each species, in addition to wild-caught specimens, three diet treatments (banana, mealworm, and starvation) were tested in triplicate. Diet-treated beetles were kept in sterile containers with sterilized soil and water for 17 days in July, 2018. This time-frame, long enough for beetles to feed 5–6 times, was chosen to examine short-term dietary impacts. Banana-fed (Trader Joe's, Dole Banana Ecuador) and mealworm-fed (Timberline, Vita-bugs Mini Mealworms 500 count) beetles were fed on the first day, and subsequently fed and watered every three days using heat-sterilized forceps and autoclaved water. All feeding portions consisted of 0.04g (+/- 0.01g) non-sterile food. Banana and mealworm bacterial communities were sequenced as controls and were removed from the analysis after confirming samples were not contaminated. Starved beetles received water, but no food. On the last day, beetles were quickly anesthetized by placing them for one minute at -80˚C in their plastic containers. All specimens, including wild-caught beetles, were dissected as described by McManus et al. [31]. Each beetle was dissected into three groups of tissues: secretory cells, gut (including foregut, midgut, and hindgut), and the rest of the body minus the secretory cells and gut (subsequently referred to as 'partial body'). Parasitic worms (Nematomorpha) found to be infecting one starved beetle and one mealworm-fed beetle were removed from those specimens and the worm tissues not included in downstream analysis.

## DNA extraction, PCR, and next generation sequencing

Tissues were incubated overnight in a 9:1 ratio of buffer ATL and proteinase K (Qiagen DNeasy Blood & Tissue Kit) at 55˚C on a rocking tray. Lysate from overnight incubation was transferred to sterile 1.5ml O-ring tubes containing 0.25g (+/- 0.02g) of 0.1mm diameter zirconium beads and bead beat at 2000rpm for 3 minutes in a PowerLyzer to lyse bacterial cells. DNA was extracted from the lysed homogenate using Solid Phase Reversible Immobilization (SPRI) magnetic beads made following the method of Rohland [32]: 100μL lysate was mixed with 180μL of well-mixed, room temperature SPRI beads, incubated for approximately 5 minutes on the bench, then transferred to a magnetic rack. After the SPRI beads pelleted, 200μL 80% ethanol was added. After 30 seconds the supernatant was removed, the ethanol wash was repeated a second time and the supernatant was removed again. Then, the tubes containing SPRI bead tubes were removed from the magnetic rack and allowed to air dry completely. DNA was eluted by adding 50μL TB solution (10mM Tris) directly onto the beads and incubating for 5 minutes, then returning samples to the magnetic rack to pellet the SPRI beads and retrieve the DNA-containing supernatant.

The V4 region of the 16S rRNA gene was PCR amplified in duplicate in 25μL reactions using GoTaq Green Master Mix (Promega), and the resulting PCR products were subsequently pooled. During the first round, previously described primers [33] 515FB_in (5′-ACA CTC TTT CCC TAC ACG ACG CTC TTC CGA TCT GTG YCA GCM GCC GCG GTA A-3′) and 806RB_in (5′-GTG ACT GGA GTT CAG ACG TGT GCT CTT CCG ATC TGG ACT ACH VGG GTW TCT AAT-3′), which were adapted to be complementary to the second round primers [34], were added to the ends of all 16S genes with the following conditions (BioRad thermocycler): initial denaturation at 94˚C for 3 min, followed by 30 cycles of 94˚C

for 45 sec, 50˚C for 1 min, 72˚C for 1:30 min, and a final extension step of 72˚C for 10 min. A second round of PCR was performed using unique combinations of barcoded forward (5'–AAT GAT ACG GCG ACC ACC GAG ATC TAC ACX XXX XXX XAC ACT CTT TCC CTA CAC GA-3') and reverse (5'-CAA GCA GAA GAC GGC ATA CGA GAT XXX XXX XXG TGA CTG GAG TTC AGA CGT G-3') primers [34] to create a dual-index amplicon library for Illumina sequencing (position of barcodes indicated by 'X' characters). The conditions for the second PCR reaction were: initial denaturation at 94˚C for 3 min, followed by 10 cycles of 94˚C for 45 sec, 50˚C for 1 min, 72˚C for 1:30 min, and a final extension step of 72˚C for 10 min. All pooled duplicate PCR products were run on a 1% agarose gel for 30 min at 100V, and imaged under UV light to verify successful PCR. DNA concentration was quantified using a Qubit fluorometer, and equimolar amounts were pooled. The pooled library was purified (Qiagen Qiaquick PCR Purification Kit) and sent for Illumina MiSeq sequencing at the U.C. Berkeley Genomics Sequencing Laboratory.

### Analysis

Amplicon reads for the V4 region of 16S were de-multiplexed with deML [35] and processed using DADA2 [36], including quality filtering with maxEE = 2. Reads were de-replicated into unique 16S amplicon sequence variants (ASVs, also referred to as phylotypes) using a read error model parameterized from the data. Paired-end reads were merged and mapped to ASVs to construct a sequence table. Chimeric sequences were removed. Taxonomic assignments for exact matches of ASVs and reference strains were made using the Ribosomal Database Project database [37]. Sequence tables and taxonomic assignments were imported into R version 3.5 [38] for downstream analysis and combined into a single phyloseq [39] object for convenience. To account for variation in sequencing effort across samples, samples were scaled according to variance stabilized ASV abundances using DESeq2 [40, 41]. ASV alignments made using DECIPHER [42, 43] were used to construct a neighbor-joining tree, and this tree was then used as the starting point for deriving a maximum likelihood tree from a generalized time-reversible model with gamma rate variation, implemented with the phangorn package in R [44]. The tree was rooted using QsRutils [45]. For comparative analysis between beetles, ASV data from all three tissues of each specimen were combined into an aggregate bacterial community. Alpha diversity measures were calculated using the packages phyloseq [39] and picante [46]. Non-metric multidimensional scaling (NMDS) plots of beta diversity were created using phyloseq [39], and analysis of similarities (ANOSIM) tests were run using the package vegan [47]. Bray-Curtis distances were calculated both for aggregate community data and for the original dataset. Venn diagrams of phylotypes present by diet were rendered by VennDiagram [48]. To control for possible sequencing errors, only phylotypes occurring at least twice in the entire dataset were included in venn diagram analysis. Hierarchical clustering of communities was performed with the package ape [49]. Secretory cells were tested for differential abundance of microbe phylotypes using an equivalent method to RNA-seq differential expression analysis, implemented using DESeq2 [39, 40].

### Ethics statement

No permits were required for the described study, which complied with all relevant regulations.

## Results

### Sequencing results

After quality filtering, the mean number of reads per sample was 19,868, and the median number of reads per sample was 17,532.

## Alpha diversity

There was a median of 95 and a mean of 98.6 ASVs present per sample. Phylogenetic diversity (PD), evenness, phylotype richness, and Shannon index were calculated for each sample.

**Diet.** PD of aggregate communities was not associated significantly with diet treatment. When tissues were considered individually, only PD of the partial body varied significantly across diet treatments (Fig 1A). Richness results were similar to PD. Neither evenness nor Shannon diversity showed any significant effect of diet in either individual tissues or aggregate microbiomes. Compared to wild-caught beetles, captivity (including both treated and control specimens) had a minor effect on PD but not on other alpha diversity measures of the microbiome. Host diet did not correlate with community alpha diversity.

**Tissue.** Tissue explained a large portion of the variance in PD of aggregate communities (Kruskal Wallis H = 55.5, p < 0.0001), so results were plotted separately for each tissue (Fig 1). Secretory cell microbiomes had higher PD than gut microbiomes (Fig 1). Richness, Shannon diversity, and evenness all varied significantly by tissue (p < 0.0001) as well. Guts had the lowest community evenness on average (average evenness 0.60, versus 0.75 in secretory cells and 0.67 in partial bodies).

**Species.** Evidence of an effect of host species on microbial community diversity was relatively weak, and varied by tissue. Overall PD and evenness did not vary significantly by species. Richness (p = 0.043) and Shannon diversity (p = 0.041) varied only slightly significantly by species. Secretory cells had a relatively consistent alpha diversity level across species, only varying significantly by the measure of evenness (p = 0.030). Gut alpha diversity varied significantly across species by richness (p = 0.027), Shannon diversity (p < 1e-05), and evenness (p < 1e-05) but not PD. Differences in gut alpha diversity appear to be driven by the exceptionally low evenness in *A*. *similis* guts. The partial body microbiome had significantly different PD (Fig 1B) and richness (p = 0.0033), but no change in evenness, across host species. PD of partial bodies was highest in *P*. *serripes*.

## Community diversity distance analysis

Bray-Curtis distances for aggregate community data (Fig 2A) and for the original dataset (Fig 2B–2D) reveal that community similarity is associated with several of the factors tested. Results of ANOSIM performed with Unifrac distances were consistent with results using Bray-Curtis distances reported below.

**Diet.** Aggregate communities did not cluster by diet (Fig 2A). They did cluster by captive (treatments as well as starved controls) versus wild-caught beetles (ANOSIM R statistic = 0.3336, p < 0.001). The only tissue that clustered significantly by diet was secretory cells, but these clustered with a lower R statistic by diet (R = 0.23) than by species (R = 0.28). Clustering by diet was explained by significant differences between captive and wild-caught beetles. Phylotypes present in aggregate communities were compared across diet treatments (Fig 3). A total of 1003 phylotypes were present across all diet conditions, 613 of which were present in wild-caught beetles. Of the phylotypes present in wild-caught beetles, 78% were present in at least one other diet condition. Just over a quarter of phylotypes were shared across all four diet conditions.

**Tissue.** Microbial communities clustered clearly and significantly by host tissue.

**Species.** Aggregate communities clustered by host species, with the *B*. *elongatulus* microbiome being the most distinct (Fig 2A). Individual tissues also clustered by species. The secretory cells had a much lower clustering statistic than the other tissues, indicating that microbial diversity in secretory cells is less differentiated than other tissues.

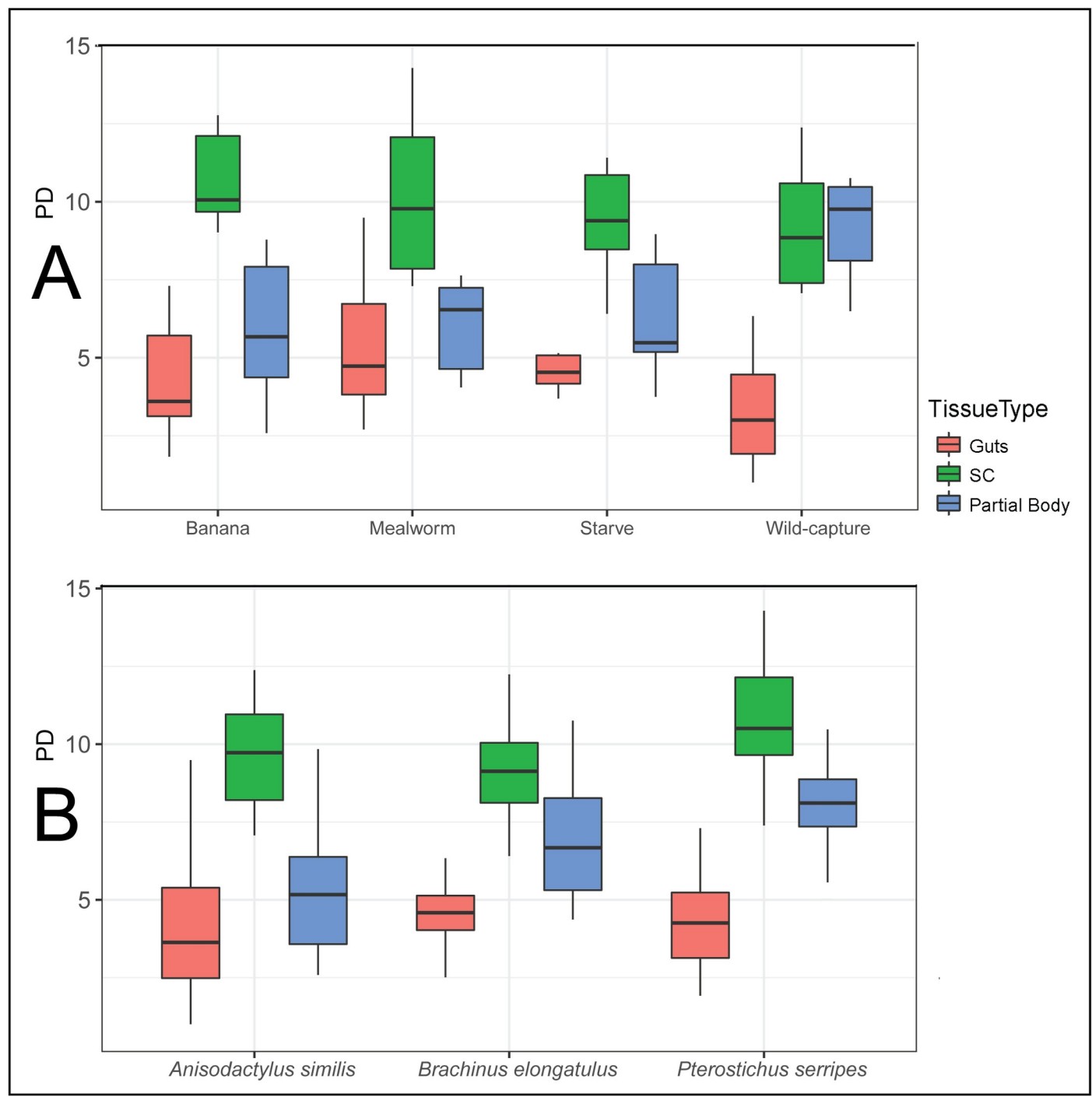

**Fig 1. Boxplots of Phylogenetic Diversity (PD), with outliers depicted as points.** (A) Plots grouped by diet treatment. PD of partial bodies varied significantly by diet treatment (H = 8.96, p = 0.030), but PD of all other tissues and of aggregate communities did not. (B) Plots grouped by host species. PD of partial bodies varied significantly by host species (Kruskal Wallis H = 8.11, p = 0.017), but PD of all other tissues and of aggregate communities did not.

## Community composition

The most abundant phyla across all samples were Proteobacteria (mean abundance 48.7%), Bacteroidetes (mean abundance 17.8%), Tenericutes, and Firmicutes. Together, these four

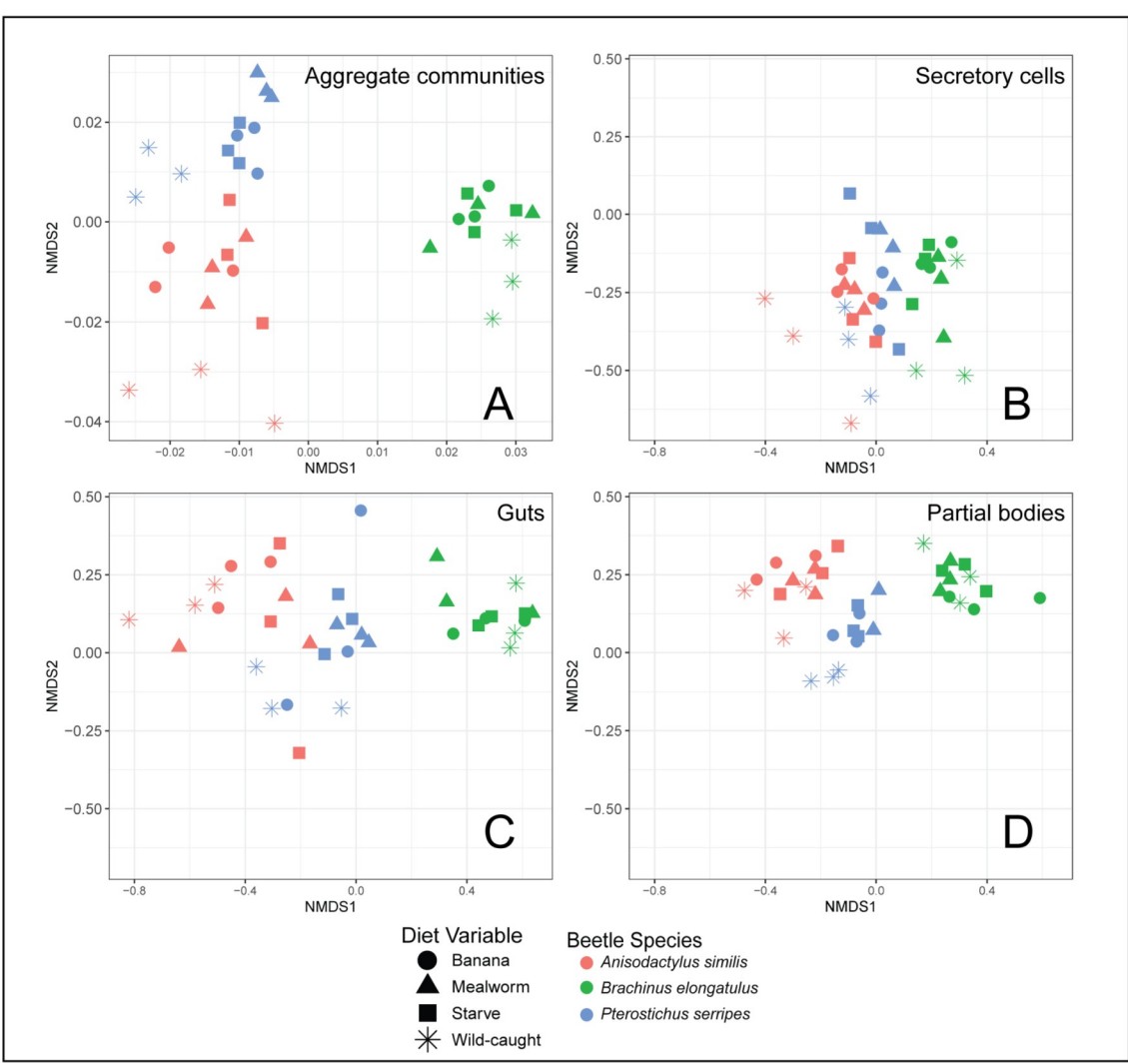

**Fig 2. Bray-Curtis ordination of microbiome beta diversity using non-metric dimensional scaling.** (A) Aggregate communities clustered significantly by species (ANOSIM R statistic = 0.92, p < 0.001) and tissue (R = 0.66, p < 0.001) only. (B) Secretory cell microbiomes clustered by species (R = 0.28, p < 0.001) and diet (R = 0.23, p < 0.001). (C) Gut microbiomes clustered clearly by species (R = 0.96, p < 0.001), and not by diet. (D) Partial body microbiomes are also clustered clearly by species (R = 0.95, p < 0.001), and not by diet.

phyla comprised a mean of 94.6% of the bacteria in each sample. Communities in all beetle species and tissues had similar phylum-level compositions. Differences by host species arose more clearly at the level of bacterial genera, so community composition of each beetle species was plotted at this level (Fig 4). Bacterial genera with median relative abundance across all samples of 1.5% or above were, in descending order of median relative abundance: *Acinetobacter*, *Spiroplasma*, *Yersinia*, *Flavobacterium*, *Pseudomonas*, *Enterobacter*, and *Enterococcus*.

**Diet.** Community composition was not significantly different across diet treatments (Fig 5).

**Tissue.** Differential abundance analysis of secretory cells versus all other tissues revealed that four phylotypes associated with two families were differentially abundant (p < 0.002). Two *Flavobacterium* phylotypes were more abundant in secretory cells than other tissues by factors of 10.22 and 16.23. Two Comamonadaceae phylotypes of unknown species were 9.38 and 9.92 fold more abundant in secretory cells. Secretory cell community composition is

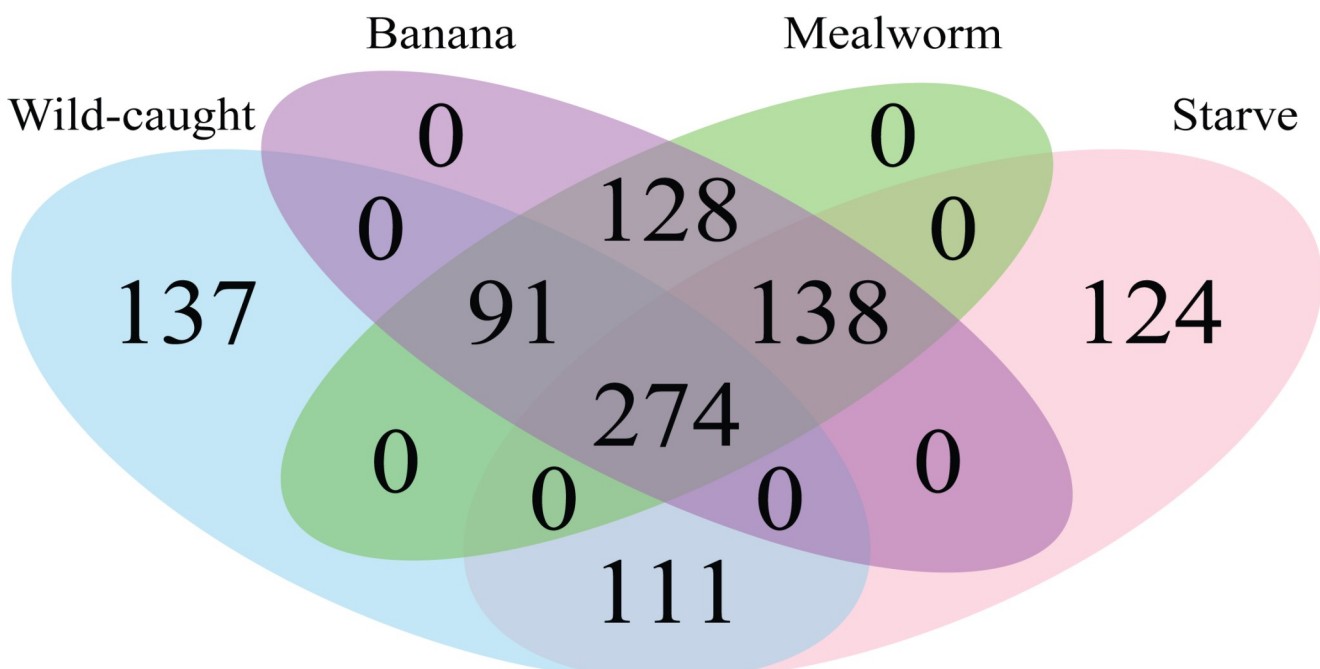

**Fig 3. Venn diagram of phylotypes present in aggregate communities by diet treatment.**

relatively conserved at the level of bacterial genera (Fig 4A). Compared to other tissues, gut microbiomes were more dominated by the ten most abundant bacterial genera; these ten genera composed over 50% of microbial abundance in all host species' guts, and over 60% of abundance in *B. elongatulus* guts (Fig 4B).

**Species.** Hierarchical clustering of community similarity showed that community differences corresponded with host species for all tissues. *Brachinus elongatulus* guts have more Firmicutes, and less Tenericutes and Actinobacteria, than the other two host species. Breaking down community composition to the genus level confirmed the status of *Brachinus* as the most distinct host species (Fig 4).

## Discussion

The present study assessed the extent to which the microbiomes of three carabid beetle species are influenced by short-term community turnover driven by diet. Sequencing gut and secretory cell communities separately provided an opportunity to examine how bacterial community composition and diversity are associated with host tissue type. We found that shifts to controlled carnivorous, herbivorous, or starvation diets had at most minor effects on bacterial species diversity or composition, regardless of host species or tissue type. In contrast, host species and tissue type explained a significant amount of the variation in microbial communities across samples. The findings of this small-scale study provide preliminary evidence that carabids harbor diverse, relatively resilient microbiomes that are persistent to short-term changes in host diet. These results contribute to broader efforts to understand host-associated microbial diversity in ground beetles.

### Microbiome resilience to host dietary change

By subjecting carabid beetles to different dietary treatments in a controlled, sterile environment, our study quantified how changes to carabid host diet influence microbiome

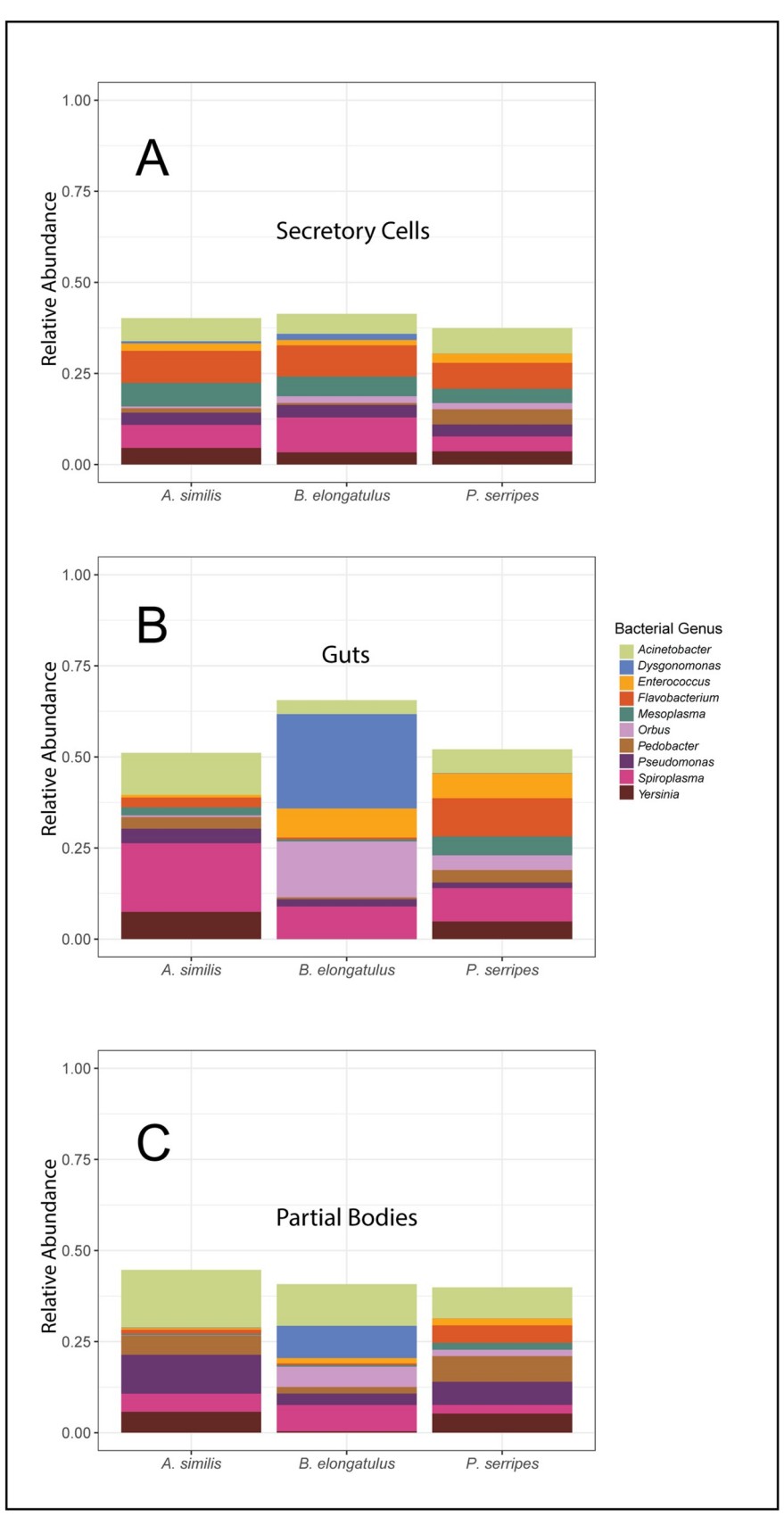

**Fig 4. Relative abundances of prevalent bacterial taxa by host tissue and species.** (A) Mean abundance in the secretory cells of the ten bacterial genera that were most abundant on average in all samples (n = 36, 12 per species). (B) Mean abundance of these bacterial genera in the guts (n = 36, 12 per species). (C) Mean abundance of these bacterial genera in the partial bodies (n = 36, 12 per species).

composition and diversity. The similarity of microbial community composition regardless of host dietary treatment indicates recently ingested food is not a primary driver of microbial community structure in these beetles. This interpretation is also supported by the finding that transient changes in host diet do not significantly alter community diversity (Fig 1). We did notice a reduction in microbial richness and diversity between wildtype beetles and all other treatment groups which were husbanded in the lab (Fig 1A), as has previously been described in carrion beetles [17] and lepidopteran species [27]. The resilience of carabid microbial communities to short-term perturbations in diet suggests that unlike caterpillars [25], carabid beetles appear to possess persistent microbial communities.

The finding that carabid microbiomes are resilient to short-term dietary shifts does not help pinpoint the factors that are most important in microbial community assembly in ground beetles. It does not address, for example, how juvenile carabids acquire microbes during their egg, larval, or pupal stages. During these life stages, microbiome acquisition and assembly may occur by selective uptake from the environment [11], vertical transmission from a parent [8], or via other routes. In addition, the nature of our study is such that it cannot distinguish whether microbiome variability is affected by long-term changes to host diet or environment. Long-term environment may affect microbial communities in carabids, as it does in *Drosophila* [11] and houseflies [24]. Future studies could investigate these and other factors that may affect carabid microbiomes throughout the carabid life cycle.

## Microbiome associations with host characteristics

Microbial communities were similar within individuals of the same host species, and distinct across the three host species. This apparent connection to host species could be a result of host defensive chemistry, host geography, vertical or horizontal transmission between individual members of the same species, environmental niche differences, or other factors. The diversity in microbiome composition we observed across host species agrees with a previous study of microbiomes in the Carabidae family which showed that the gut microbiomes of two species of carabids, *Harpalus pensylvanicus* (Degeer, 1774) and *Anisodactylus sanctaecrucis* (Fabricius, 1798), have different composition and species richness from each other [50]. The prevalence of the genus *Spiroplasma* in our results agrees with the findings of previous studies of carabid microbiomes [31, 50]. *Dysgonomonas*, which has previously been found to be prevalent in *B. elongatulus* [31], was one of the most abundant genera, and also had higher relative abundance in *B. elongatulus* than in other beetle species, especially in the guts (Fig 3). *Enterococcus*, which was previously found in the digestive tract of *B. elongatulus* [31], was again found in that species and also in *P. serripes*, both at a greater abundance in the guts than in other tissues (Fig 3). Although microbiome community structure is often not correlated with insect host phylogeny at higher taxonomic level such as order [11, 21], we note that some of our results align with previous findings in the order Coleoptera. Specifically, several highly abundant bacterial genera in our samples were previously found in *Cephaloleia* (Chrysomelidae) beetles [14]. The phylogenetic diversity of these carabid microbiomes is also within the compass of previous studies in Coleoptera [23].

We believe our study to be the first to comparatively describe microbial communities found specifically in the pygidial gland secretory cells, a functionally important tissue with

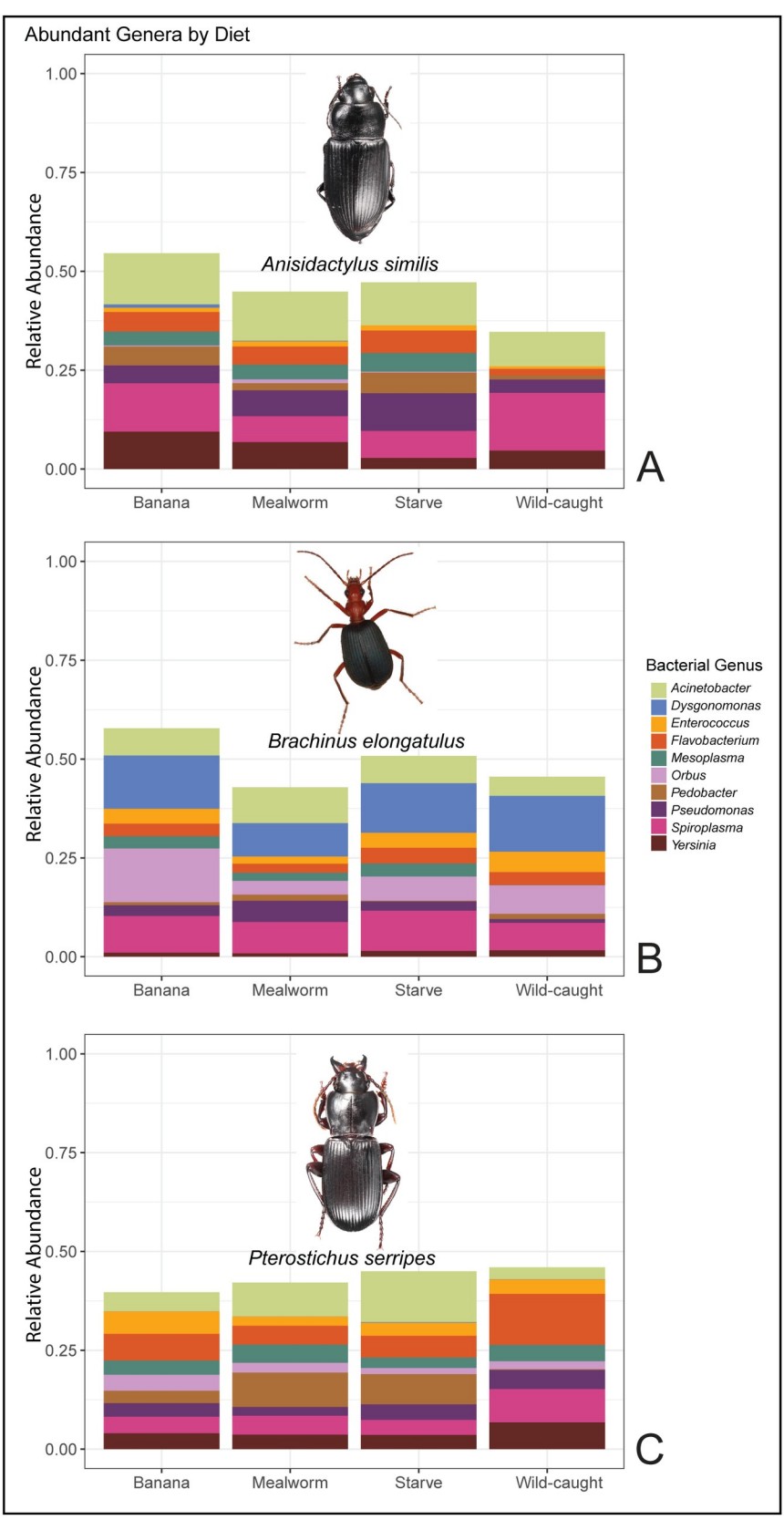

**Fig 5. Relative abundances of prevalent bacterial taxa by host diet treatment.** (A) Mean abundance in *A. similis* of the ten most abundant bacterial genera across all samples (n = 36), grouped by diet treatment (n = 9, each). (B) Mean abundance in *B. elongatulus* of these bacterial genera, grouped by diet treatment. (C) Mean abundance of these bacterial genera in *P. serripes*, grouped by diet treatment. Genera included are the same as in Fig 4. Photographs of beetles depict typical host morphology.

homologous structures found in carabids and other Adephaga. Our preliminary study found that the pygidial glands possess a differentiated microbiome from other carabid tissues, warranting further investigation especially given the unique metabolic capabilities of these cells. In our samples, the phylogenetic diversity of microbial communities in the secretory cells of these glands is particularly high (Fig 1). The composition of secretory cell communities is more similar across host species than that of other tissues (Figs 2B and 4). Secretory cells were also the only tissue to see a significant change in their microbial community in response to diet treatment. This finding is notable since guts would presumably be the tissue most closely associated with diet. It is tempting to speculate that changes to metabolic inputs that come with dietary shifts deprive secretory cell bacteria of required substrates, but a larger, more targeted experimental study would be required to understand how secretory cell bacteria and metabolic activity respond to host diet. As a baseline for future studies of carabid secretory cell microbes to build upon, we determined that several bacterial genera in particular (*Flavobacterium* and an unknown Comamonadaceae genus) are differentially abundant in the secretory cells over other tissues. In addition to more thorough investigations of secretory cell microbiome composition across Carabidae, we propose that future studies should directly test the possibility that a symbiotic relationship with microbes plays a role in host chemical biosynthesis. This could be done in several ways, such as by using antibiotics to flush beetles of their microbes, or by experimentally confirming the metabolic activity of bacterial isolates from secretory cells.

The present study found a strong association between host tissue type and microbiome characteristics, not only in secretory cells, but also in guts. Out of all the factors controlled for in this study, tissue type (secretory cells, guts, or partial bodies) has the closest association with microbiome composition and diversity. Tissue identity explains much of the variation between communities, in composition and distance ordination, and most of the variation in PD (Fig 1). Previous research has found that factors such as environmental filtering [3] and routes of microbe dispersal [22] can shape microbiome composition, and the especially strong association with tissue type could be related to these factors. Future efforts to assess functional microbe-host interactions, such as possible connections between gut microbes and carabid host nutrition, might consider directly quantifying functional genes and metabolic pathways present in the microbial communities.

## Acknowledgments

We thank Wendy Moore and her lab group at the University of Arizona, Tucson, for providing the *Brachinus elongatulus* for the study.

## Author Contributions

**Conceptualization:** Kipling Will, Aman Gill.

**Data curation:** Aman Gill.

**Formal analysis:** Anita Silver, Aman Gill.

**Funding acquisition:** Kipling Will.

**Investigation:** Sean Perez, Melanie Gee, Bethany Xu, Shreeya Garg.

**Methodology:** Sean Perez, Aman Gill.

**Resources:** Kipling Will.

**Supervision:** Kipling Will, Aman Gill.

**Visualization:** Anita Silver.

**Writing – original draft:** Anita Silver.

**Writing – review & editing:** Anita Silver, Sean Perez, Melanie Gee, Bethany Xu, Shreeya Garg, Kipling Will, Aman Gill.

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
