## [Decision Letter · Decision Letter 0]

30 Dec 2020

PONE-D-20-32322

Persistence of the ground beetle (Coleoptera: Carabidae) microbiome to diet manipulation

PLOS ONE

Dear Dr. Kipling

Thank you for submitting your manuscript to PLOS ONE. After careful consideration, we feel that it has merit but does not fully meet PLOS ONE’s publication criteria as it currently stands. Therefore, we invite you to submit a revised version of the manuscript that addresses the points raised during the review process.

Both reviewers found the manuscript interesting, but have recomended substantial modifications. Please pay special attention to the comment by reviewr #2 regarding the sample sizes.

We look forward to receiving your revised manuscript.

Kind regards,

Omri Finkel, PhD

Academic Editor

PLOS ONE

Journal Requirements:

Reviewers' comments:

Reviewer's Responses to Questions

**Comments to the Author**

1. Is the manuscript technically sound, and do the data support the conclusions?

Reviewer #1: Partly

Reviewer #2: Partly

2. Has the statistical analysis been performed appropriately and rigorously? 

Reviewer #1: Yes

Reviewer #2: I Don't Know

3. Have the authors made all data underlying the findings in their manuscript fully available?

Reviewer #1: Yes

Reviewer #2: Yes

4. Is the manuscript presented in an intelligible fashion and written in standard English?

Reviewer #1: Yes

Reviewer #2: Yes

5. Review Comments to the Author

Reviewer #1: Summary

The manuscript investigates the bacterial community of three species of carabid beetles, evaluating the differences according to insect host, tissue, diet and environment. The dataset is valuable, as there is otherwise scarce research on the microbial communities associated to this beetle family. The consideration of different diets and tissues is additionally relevant. However, from my view there is an important methodological issue that should be addressed, and which might confound the interpretation of the data. Also, the way in which some of the questions are posed, as well as some of the arguments used in the discussion can be misleading. I point out specific issues and suggest some improvements to better align the conclusions with the approach and observations.

General comments

1. In lines 117-118, the authors put forward an aim consistent with the experiments that were carried out, i.e. “This study examined how the transient factor of diet treatment, and more permanent factors including host species and tissue, contribute to the observed variation in carabid associated microbiomes”. However, later in Lines 124-131 three predictions are put forward in relation to whether the communities are non-transient and to potential functions. This, in my opinion, generates confusion as to what can actually be deduced from the data, specially given the experimental approach. To test for non-transient communities, a broader sampling from different life stages and/or across individuals, or at least placing focus on individual variation would be appropriate. I suggest that the authors remain by the narrative in lines 117-118 and are more cautious in the interpretations, as expanded on in several of the next comments.

2. There is an important confounding factor in the methodology: given that the individuals were not cleared of their original microbiota, it is rather expected that the existing communities are not easily replaced. Another very important aspect is the original establishment of the microbial community, which is likely in earlier life stages. Although the authors do bring this point into discussion, I believe there should be much more emphasis on the relevance of the originally established community and the factors that determine that. These are likely confounding factors, which could be tied to host species and tissue, and thus explain the observed results.

3. Also related to the previous point: on what basis was 17 days chosen as an evaluation period? It might indeed be hard to know which duration is most appropriate to see potential changes, but additional information is useful: How long do these beetles spend in their adult stage? How often do they feed?

4. Associating the (partially) characteristic bacterial composition according to host tissue to potential function is rather weakly founded. Authors somehow acknowledge it, but the overall approach of trying to tie function to this kind of analysis is loose /weak (see specific comment for lines 129-131 and line 347). Also, the fact that the sample size from natural conditions are low make it hard to conclude on consistency, which is likely important for function.

5. I suggest reconsidering the conclusion that: “Symbiosis is a possibility, especially in secretory cells” (see specific comment on lines 454).

6. Presentation of the data in the figures and the references to it in the text can be considerably improved. It was hard to relate to each figure (they are not in order and panel letters not always given). The figures can also become more informative, please see specific comments.

7. The methods section, as well as the background information provided in the introduction are clearly written and structured.

Specific comments:

- Lines 25-26: both here and in the discussion (line 418), the “typical” microbial diversity found in “other insect hosts” is rather ambiguous, since there is a relatively broad spectrum of community types, richness and diversity levels across insect orders, and even within orders. Please instead refer to closer groups for comparison.

- Lines 29-31: as mentioned in the general comments, this conclusion is not supported by the data. There is a significant difference in the bacterial community composition in secretory cells according to species (Fig 2b + legend). Also, if the defensive chemistry is different between species (Line 21), it would also not be coherent to expect highly similar communities if they are involved in producing the defensive compounds.

- Lines 77-79: This is not necessarily true. The set of metabolic pathways of two communities with distinct microbial compositions can in some cases match to a good extent. Different bacteria can play similar roles, the same holds for groups of bacteria.

- Line 124: the (1) for numbering the predictions should be placed in line 125, at “…, then 1) diet..” But see next comment.

- Lines 124 – 125: how this is written suggests that the question addressed is whether carabids harbor non-transient microbes. The experimental approach does not address this though (see general comment 1).

- Lines 129-131: True to some extent, but the link to the question of function is quite loose. Especially considering the contrary situation (microbial communities are not random in respect to tissue), since this is still far from indicating function, specially in rather complex microbial communities like those in carabids.

- Lines 147-149: Does this mean that the diets were completely sterile? Please indicate explicitly.

- Line 226: Here and throughout the manuscript, please indicate the specific panel of the figure referred to (e.g. Fig 1B). Also, please reorder to follow the sequential order of the figure numbers.

- Lines 229-230: I suggest mentioning here or in the methods the list of alpha diversity measures that were evaluated.

- Lines 241: I consider it useful to provide these results here or as a supplement.

- Line 307: the four phylotypes hare present in different abundance, not differentially expressed. Please correct.

- Lines 311-314: this is probably easier to express and understand in the context of evenness. This observation and its importance are hard to grasp as currently stated.

- Lines 330-333: as mentioned in the general comments, the experimental setup involved a relatively short period of time and used beetles with an already established microbiota. Thus, I suggest toning down this statement “these findings demonstrate that carabid microbiomes are highly persistent…”

- Line 347: The hypothesis is supported, but the composition in specific tissues is by no means evidence of function. Because the result can be explained by the different physicochemical conditions in each tissue, or the exposure of certain tissues to different sources of microbes (as also mentioned by the authors), this should not be set as a strong proxy for function, but rather mentioned as a side observation which can be addressed differently.

- Lines 354-355: I disagree with this conclusion based on general comment 2. Please reconsider.

- Lines 358-360: this is different to the 2nd hypothesis mentioned in the introduction, or what are the authors referring to? Also, is there a reason to believe that they transmit them between conspecifics? Gregarious or social behavior would be more in line with such hypothesis, not the fact of seeing similarities across individuals. This could also be the case for vertically transmitted symbionts.

- Line 361: referring to co-evolution is a long stretch in this system. Co-evolution would be expected if there is consistent vertical transmission, but there is no evidence for that here, and it is also not the appropriate data set to address this question.

- Lines 404-407: it was not clear to me what the authors mean and how this connects to the previous lines. Please revise.

- Lines 415-416: I believe this is not a fair claim. While the study is valuable for learning about these 3 insect species, these are only 3 of the most diverse group of animals.

- Line 444: does this match the fact that they were found in all species or a specific species? i.e. are quinones produced in all hosts?

- Line 454: I don’t think that this an appropriate interpretation of the data, at least not based on the results shown. If the definition of symbiosis is used: “the consistent association of individuals from different species for all or most of their lifetime”. If the figures provide support that the communities within each species are consistent across individuals from different populations or collection sites, then this conclusion is better supported. The results from the secretory glands just show some degree of convergence in composition, but this is not per se evidence for symbiosis.

- Figure 4. This figure could become more informative and easier to grasp by

o Showing the composition of each replicate, to give a better impression of the amount of variation per individual.

o Separating or arranging the panels to address a single factor (species or tissue) and therefore message per figure

o In panel 4D, it is not so clear why the species are merged if Fig. 2 already shows that there is clear clustering of composition per species within each body part. I suggest separating this per species, and (in line with the point above), make it a separate figure.

- Figures 4 and 5: I suggest labelling the y-axis as “Relative abundance” for clarity.

Reviewer #2: This paper reports carabid associated microbiomes. There are 3 species, 4 diet groups (3 lab, 1 field), and 3 tissues examined (includes partial body). The hypotheses involve development of microbiomes that are specific to beetle species, their chemical defense capabilities, and possible role in evolution of these capabilities.

It is obvious that the authors have conducted a lot of research in this area and are familiar with related literature. The writing is generally clear, well-organized, and understandable. The abstract should be restructured – findings (lines 19-20) are ahead of methods. No mention in the Abstract of gut microbiota across the 3 species, just pygidial gland. Intro, Line 78 – “they” = carabids (right?). Perhaps restructure sentence to make clear.

The biggest problem I have is the n=3 beetles for each of the treatment groups (12) wherein a treatment group is defined by the carabid species (3) and diet treatment (4 including wild). This is OK for a preliminary study, but not a full-fledged study that would support robust conclusions. If this paper is to be published with such a low n, then the writing needs to clearly convey that these are preliminary findings, reduce the level of confidence in concluding statements, and shorten the Discussion considerably.

The second issue is that “soil” or “environment” are not tested in this study, and mention of such needs to be eliminated from the manuscript (see lines 16, 24, 119, 125, 321, 323, 327, 339, 344, etc). Three wild caught beetles do not compare soil or environment with three groups of beetles lab-reared on 3 different diets.

The third issue is the statistics. Is Chi-Square the right analysis for this study? Can you give more details of what data was used for the chi-sq? Was is it a binary presence absence of bacterial taxa or just a single diversity metric? How many taxa were considered? What were the p values for the non-significant tests? When I look at figure 1, I see tissue types have different diversity. And Wild capture partial body maybe different than lab diets. That all makes sense. Beyond that, I don’t know. Figure 2 suggest differences between carabid species for each tissue microbiome. with secretory cells having the least range across species. And diet having little effect. But remember in this type of analysis, its all relative. If you just isolated diet for a single species, you might see distinct clustering there. I’m unconvinced that chi-square can answer to hypotheses proposed in lines 124-131. Did you test randomness by other tests (lines 119-121)? What did you used to explain variation in microbiomes?

6. PLOS authors have the option to publish the peer review history of their article (what does this mean?). If published, this will include your full peer review and any attached files.

Reviewer #1: **Yes: **Laura V. Florez

Reviewer #2: No

---

## [Author Response · Author response to Decision Letter 0]

20 Feb 2021

We would like to thank the reviewers for their thoughtful and thorough comments regarding our manuscript, entitled, “Persistence of the ground beetle (Coleoptera: Carabidae) microbiome to diet manipulation.” We have addressed each comment individually, below. We believe these changes have significantly strengthened our paper, and we hope this revised version satisfies the publication standards of PLOS ONE. 

1. Is the manuscript technically sound, and do the data support the conclusions?

Reviewer #1: Partly

Reviewer #2: Partly

Thank you for your feedback. 

2. Has the statistical analysis been performed appropriately and rigorously?

Reviewer #1: Yes

Reviewer #2: I Don't Know

Thank you for your feedback. 

3. Have the authors made all data underlying the findings in their manuscript fully available?

Reviewer #1: Yes

Reviewer #2: Yes

Thank you.

4. Is the manuscript presented in an intelligible fashion and written in standard English?

Reviewer #1: Yes

Reviewer #2: Yes

Thank you. 

5. Review Comments to the Author

Reviewer #1: 

Summary

The manuscript investigates the bacterial community of three species of carabid beetles, evaluating the differences according to insect host, tissue, diet and environment. The dataset is valuable, as there is otherwise scarce research on the microbial communities associated to this beetle family. The consideration of different diets and tissues is additionally relevant. However, from my view there is an important methodological issue that should be addressed, and which might confound the interpretation of the data. Also, the way in which some of the questions are posed, as well as some of the arguments used in the discussion can be misleading. I point out specific issues and suggest some improvements to better align the conclusions with the approach and observations.

Thank you for your summary of our study and its importance. We appreciate your suggestions of how to better align our discussion with the data which we collected. 

General comments

1. In lines 117-118, the authors put forward an aim consistent with the experiments that were carried out, i.e. “This study examined how the transient factor of diet treatment, and more permanent factors including host species and tissue, contribute to the observed variation in carabid associated microbiomes”. However, later in Lines 124-131 three predictions are put forward in relation to whether the communities are non-transient and to potential functions. This, in my opinion, generates confusion as to what can actually be deduced from the data, specially given the experimental approach. To test for non-transient communities, a broader sampling from different life stages and/or across individuals, or at least placing focus on individual variation would be appropriate. I suggest that the authors remain by the narrative in lines 117-118 and are more cautious in the interpretations, as expanded on in several of the next comments.

Thank you for this helpful suggestion. We have completely rewritten the last paragraph of our introduction to better fit the type of conclusions that can be drawn from this experimental approach. We have also emphasized throughout the paper that we are only interested in distinguishing whether communities experience significant turn-over of species as a result of short-term perturbations. 

2. There is an important confounding factor in the methodology: given that the individuals were not cleared of their original microbiota, it is rather expected that the existing communities are not easily replaced. Another very important aspect is the original establishment of the microbial community, which is likely in earlier life stages. Although the authors do bring this point into discussion, I believe there should be much more emphasis on the relevance of the originally established community and the factors that determine that. These are likely confounding factors, which could be tied to host species and tissue, and thus explain the observed results.

This is an excellent point, and we agree that our study is not able to disentangle confounding factors that might be relevant in the establishment of a host-associated microbiome. Instead, our aim was to establish whether pre-existing communities were in place and determine the degree of persistence of these microbial communities to short term diet changes. We believe the extensive revisions we have made to the discussion section have helped to emphasize this. 

3. Also related to the previous point: on what basis was 17 days chosen as an evaluation period? It might indeed be hard to know which duration is most appropriate to see potential changes, but additional information is useful: How long do these beetles spend in their adult stage? How often do they feed?

Based on this helpful comment, we have added a sentence to the methods section providing reasoning for the choice of a 17-day period. 

4. Associating the (partially) characteristic bacterial composition according to host tissue to potential function is rather weakly founded. Authors somehow acknowledge it, but the overall approach of trying to tie function to this kind of analysis is loose /weak (see specific comment for lines 129-131 and line 347). Also, the fact that the sample size from natural conditions are low make it hard to conclude on consistency, which is likely important for function.

This is an important point. As part of our revisions, we have edited passages which previously implied that our data demonstrates microbial community involvement with host functions. 

5. I suggest reconsidering the conclusion that: “Symbiosis is a possibility, especially in secretory cells” (see specific comment on lines 454).

Thank you for the suggestion. As stated above, we decided based on your feedback to cut that conclusion from our revised manuscript. 

6. Presentation of the data in the figures and the references to it in the text can be considerably improved. It was hard to relate to each figure (they are not in order and panel letters not always given). The figures can also become more informative, please see specific comments.

This is helpful to know, and we have made several changes to this end. 

7. The methods section, as well as the background information provided in the introduction are clearly written and structured.

Thank you. 

Specific comments:

- Lines 25-26: both here and in the discussion (line 418), the “typical” microbial diversity found in “other insect hosts” is rather ambiguous, since there is a relatively broad spectrum of community types, richness and diversity levels across insect orders, and even within orders. Please instead refer to closer groups for comparison.

Based on your feedback, we have removed comparisons across Insecta where they occur in the Abstract and also in the Discussion. 

- Lines 29-31: as mentioned in the general comments, this conclusion is not supported by the data. There is a significant difference in the bacterial community composition in secretory cells according to species (Fig 2b + legend). Also, if the defensive chemistry is different between species (Line 21), it would also not be coherent to expect highly similar communities if they are involved in producing the defensive compounds.

We have cut this sentence from the paper, and softened all conclusions based upon secretory cell similarity. 

- Lines 77-79: This is not necessarily true. The set of metabolic pathways of two communities with distinct microbial compositions can in some cases match to a good extent. Different bacteria can play similar roles, the same holds for groups of bacteria.

We have rewritten this sentence to take a more tentative stance about this possibility. 

- Line 124: the (1) for numbering the predictions should be placed in line 125, at “…, then 1) diet..” But see next comment.

Thank you. We have made this change.

- Lines 124 – 125: how this is written suggests that the question addressed is whether carabids harbor non-transient microbes. The experimental approach does not address this though (see general comment 1).

We have revised the paper significantly to address this concern, as described under general comment 1, and throughout the highlighted sections of the revised manuscript. We have also rewritten this last paragraph of the introduction to better align with the aims and scope of our experiments. 

- Lines 129-131: True to some extent, but the link to the question of function is quite loose. Especially considering the contrary situation (microbial communities are not random in respect to tissue), since this is still far from indicating function, specially in rather complex microbial communities like those in carabids.

We have made this sentence more preliminary and less definitive. 

- Lines 147-149: Does this mean that the diets were completely sterile? Please indicate explicitly.

We have added, “All feeding portions consisted of 0.04g (+/- 0.01g) non-sterile food.”

- Line 226: Here and throughout the manuscript, please indicate the specific panel of the figure referred to (e.g. Fig 1B). Also, please reorder to follow the sequential order of the figure numbers.

We have specified panels in most figure references. 

- Lines 229-230: I suggest mentioning here or in the methods the list of alpha diversity measures that were evaluated.

The list of alpha diversity measures has been added. 

- Lines 241: I consider it useful to provide these results here or as a supplement.

We appreciate this suggestion, but we felt that the limited scope of this study did not warrant creating a supplementary file. 

- Line 307: the four phylotypes hare present in different abundance, not differentially expressed. Please correct.

Thank you for catching this error. It has been corrected. 

- Lines 311-314: this is probably easier to express and understand in the context of evenness. This observation and its importance are hard to grasp as currently stated.

We have added a sentence earlier in the results section to note that the evenness of guts is lower than that of other tissues, across samples. We have also kept this observation in the paper to provide additional detail. 

- Lines 330-333: as mentioned in the general comments, the experimental setup involved a relatively short period of time and used beetles with an already established microbiota. Thus, I suggest toning down this statement “these findings demonstrate that carabid microbiomes are highly persistent…”

We have toned down this passage by rewriting it as follows:

The findings of this small-scale study provide preliminary evidence that carabids harbor diverse, relatively resilient microbiomes that are persistent to short-term changes in host diet. These results contribute to broader efforts to understand host-associated microbial diversity in ground beetles. 

- Line 347: The hypothesis is supported, but the composition in specific tissues is by no means evidence of function. Because the result can be explained by the different physicochemical conditions in each tissue, or the exposure of certain tissues to different sources of microbes (as also mentioned by the authors), this should not be set as a strong proxy for function, but rather mentioned as a side observation which can be addressed differently.

We have modified the manuscript so that it is clear that associations may be due to functional relationships or one of various other reasons. The emphasis on function has been reduced to some brief passages noting associations, as suggested. 

- Lines 354-355: I disagree with this conclusion based on general comment 2. Please reconsider.

As part of significant re-organization and editing of the discussion, we have removed this sentence.

- Lines 358-360: this is different to the 2nd hypothesis mentioned in the introduction, or what are the authors referring to? Also, is there a reason to believe that they transmit them between conspecifics? Gregarious or social behavior would be more in line with such hypothesis, not the fact of seeing similarities across individuals. This could also be the case for vertically transmitted symbionts.

As part of significant re-organization and editing of the discussion, we have removed this sentence. In its place, we briefly mention several possible reasons why microbial communities may be similar in individuals of the same species. 

- Line 361: referring to co-evolution is a long stretch in this system. Co-evolution would be expected if there is consistent vertical transmission, but there is no evidence for that here, and it is also not the appropriate data set to address this question.

This conjecture has been removed from the manuscript, thank you for the feedback. 

- Lines 404-407: it was not clear to me what the authors mean and how this connects to the previous lines. Please revise.

This section of the discussion has been rearranged, and the problematic passage has been clarified so that it is easier to follow and understand. 

- Lines 415-416: I believe this is not a fair claim. While the study is valuable for learning about these 3 insect species, these are only 3 of the most diverse group of animals.

In our efforts to make the discussion section shorter and more concise, we have removed the paragraph that compares our results to others across Insecta. 

- Line 444: does this match the fact that they were found in all species or a specific species? i.e. are quinones produced in all hosts?

We have removed this sentence, and so the comment no longer applies. 

- Line 454: I don’t think that this an appropriate interpretation of the data, at least not based on the results shown. If the definition of symbiosis is used: “the consistent association of individuals from different species for all or most of their lifetime”. If the figures provide support that the communities within each species are consistent across individuals from different populations or collection sites, then this conclusion is better supported. The results from the secretory glands just show some degree of convergence in composition, but this is not per se evidence for symbiosis.

We have removed this conclusion based on your general comments, so this comment no longer applies. 

- Figure 4. This figure could become more informative and easier to grasp by

o Showing the composition of each replicate, to give a better impression of the amount of variation per individual.o Separating or arranging the panels to address a single factor (species or tissue) and therefore message per figure

This would be a helpful view but we felt that the ordination plots show that compositional differences among replicates within groups are not large compared to variation between groups, so for simplicity we retained the averaged plots.

o In panel 4D, it is not so clear why the species are merged if Fig. 2 already shows that there is clear clustering of composition per species within each body part. I suggest separating this per species, and (in line with the point above), make it a separate figure.

We have removed figure 4D, since upon reviewing the results and discussion we felt that it did not make an important contribution that was worthy inclusion as a separate figure. 

- Figures 4 and 5: I suggest labelling the y-axis as “Relative abundance” for clarity.

We have made this revision. 

Reviewer #2:

This paper reports carabid associated microbiomes. There are 3 species, 4 diet groups (3 lab, 1 field), and 3 tissues examined (includes partial body). The hypotheses involve development of microbiomes that are specific to beetle species, their chemical defense capabilities, and possible role in evolution of these capabilities.

Thank you for your clear and concise summary of our work.

It is obvious that the authors have conducted a lot of research in this area and are familiar with related literature. The writing is generally clear, well-organized, and understandable. 

Thank you. 

The abstract should be restructured – findings (lines 19-20) are ahead of methods. No mention in the Abstract of gut microbiota across the 3 species, just pygidial gland. Intro, Line 78 – “they” = carabids (right?). Perhaps restructure sentence to make clear.

Thank you for pointing this out. We have revised the abstract. All results are described after methods, and our findings are described more clearly. 

The biggest problem I have is the n=3 beetles for each of the treatment groups (12) wherein a treatment group is defined by the carabid species (3) and diet treatment (4 including wild). This is OK for a preliminary study, but not a full-fledged study that would support robust conclusions. If this paper is to be published with such a low n, then the writing needs to clearly convey that these are preliminary findings, reduce the level of confidence in concluding statements, and shorten the Discussion considerably.

We appreciate this feedback, and have taken several steps to improve our manuscript based on it. We have completely re-written the scope of hypotheses we are intending to address to better fit with our dataset, modified the language of the manuscript to be more tentative throughout, and shortened the Discussion section significantly. We have also added a brief nod to the low sample size in the Methods section. 

The second issue is that “soil” or “environment” are not tested in this study, and mention of such needs to be eliminated from the manuscript (see lines 16, 24, 119, 125, 321, 323, 327, 339, 344, etc). Three wild caught beetles do not compare soil or environment with three groups of beetles lab-reared on 3 different diets.

We realize that this choice of wording was somewhat misleading, and we have revised the manuscript to make our actual results more clear.

The third issue is the statistics. Is Chi-Square the right analysis for this study? Can you give more details of what data was used for the chi-sq? 

The specific test used was Kruskal-Wallis, and we have corrected references of “chi-squared” to “H”, as this is the correct term for the Kruskal-Wallis statistic. We used this statistic because it is a common way for performing a non-parametric comparison between multiple groups. 

Was is it a binary presence absence of bacterial taxa or just a single diversity metric? 

Each diversity metric we used (richness, evenness, Shannon index, phylogenetic diversity) takes different factors (taxa presence/absence, relative abundance, etc.) into account in a slightly different way. Phylogenetic diversity considers not only which bacterial taxa are present, but also their degree of phylogenetic relatedness to each other. 

How many taxa were considered? 

All bacterial taxa were included in the metric. 

What were the p values for the non-significant tests? 

The p-values for non-significant results referenced in figure captions are as follows:

Figure panel in which the test is referenced Value Group (independent variable) p-value

Figure 1A PD of guts Host species 0.70

Figure 1A PD of secretory cells Host species 0.16

Figure 1A PD of aggregate communities Host species 0.27

Figure 1B PD of guts Diet treatment 0.09

Figure 1B PD of secretory cells Diet treatment 0.49

Figure 1B PD of aggregate communities Diet treatment 0.99

Figure 2A Bray-Curtis clustering of aggregate communities Diet treatment 0.06

Figure 2C Bray-Curtis clustering of guts Diet treatment 0.67

Figure 2D Bray-Curtis clustering of partial bodies Diet treatment 0.09

When I look at figure 1, I see tissue types have different diversity. And Wild capture partial body maybe different than lab diets. That all makes sense. 

Thank you. 

Beyond that, I don’t know. Figure 2 suggest differences between carabid species for each tissue microbiome. with secretory cells having the least range across species. And diet having little effect. But remember in this type of analysis, its all relative. 

Thank you for pointing this out. 

If you just isolated diet for a single species, you might see distinct clustering there. I’m unconvinced that chi-square can answer to hypotheses proposed in lines 124-131. 

This is a fair point. We believe it has been addressed by the significant changes we have made in our manuscript, and in particular by the substantial changes we have made to the end of the Introduction, which clarify what our study is able to accomplish. 

Did you test randomness by other tests (lines 119-121)? 

Based on your feedback, we have changed the way we write about our hypotheses and findings to be clearer and more accurate (instead of generically referring to ‘variation’). We believe that the edited manuscript has addressed this comment.

What did you used to explain variation in microbiomes?

We defined variation between communities as significant differences in their community diversity and composition. Our study tested several factors that might potentially explain such variation.

---

## [Decision Letter · Decision Letter 1]

4 Mar 2021

Persistence of the ground beetle (Coleoptera: Carabidae) microbiome to diet manipulation

PONE-D-20-32322R1

Dear Dr. Will,

We’re pleased to inform you that your manuscript has been judged scientifically suitable for publication and will be formally accepted for publication once it meets all outstanding technical requirements.

Kind regards,

Omri Finkel, PhD

Academic Editor

PLOS ONE

Additional Editor Comments (optional):

Reviewers' comments:

Reviewer's Responses to Questions

**Comments to the Author**

1. If the authors have adequately addressed your comments raised in a previous round of review and you feel that this manuscript is now acceptable for publication, you may indicate that here to bypass the “Comments to the Author” section, enter your conflict of interest statement in the “Confidential to Editor” section, and submit your "Accept" recommendation.

Reviewer #1: All comments have been addressed

Reviewer #2: All comments have been addressed

2. Is the manuscript technically sound, and do the data support the conclusions?

Reviewer #1: Yes

Reviewer #2: (No Response)

3. Has the statistical analysis been performed appropriately and rigorously? 

Reviewer #1: Yes

Reviewer #2: (No Response)

4. Have the authors made all data underlying the findings in their manuscript fully available?

Reviewer #1: Yes

Reviewer #2: (No Response)

5. Is the manuscript presented in an intelligible fashion and written in standard English?

Reviewer #1: Yes

Reviewer #2: (No Response)

6. Review Comments to the Author

Reviewer #1: (No Response)

Reviewer #2: (No Response)

7. PLOS authors have the option to publish the peer review history of their article (what does this mean?). If published, this will include your full peer review and any attached files.

Reviewer #1: **Yes: **Dr. Laura V. Florez

Reviewer #2: No

---

## [Editor Report · Acceptance letter]

12 Mar 2021

PONE-D-20-32322R1 

Persistence of the ground beetle (Coleoptera: Carabidae) microbiome to diet manipulation 

Dear Dr. Will:

I'm pleased to inform you that your manuscript has been deemed suitable for publication in PLOS ONE. Congratulations! Your manuscript is now with our production department. 

Kind regards, 

on behalf of

Dr. Omri Finkel 

Academic Editor

PLOS ONE